# Enhanced protection face masks do not adversely impact thermophysiological comfort

**Farzan Gholamreza[1], Anupama Vijaya Nadaraja[1], Abbas S. Milani[1], Kevin Golovin [1,2]***

**1** School of Engineering, University of British Columbia, Kelowna, Canada, **2** Department of Mechanical & Industrial Engineering, University of Toronto, Toronto, Canada

* kevin.golovin@utoronto.ca

## Abstract

The World Health Organization has advocated mandatory face mask usage to combat the spread of COVID-19, with multilayer masks recommended for enhanced protection. However, this recommendation has not been widely adopted, with noncompliant persons citing discomfort during prolonged usage of face masks. And yet, a scientific understanding on how face mask fabrics/garment systems affect thermophysiological comfort remains lacking. We aimed to investigate how fabric/garment properties alter the thermal and evaporative resistances responsible for thermophysiological strain. We constructed 12 different layered facemasks (D1-D5, T1-T6, Q1) with various filters using commercially available fabrics. Three approaches were employed: (1) the evaporative and thermal resistances were measured in all the test face masks using the medium size to determine the effect of fabric properties; (2) the effect of face mask size by testing close-fitted (small), fitted (medium) and loose fitted (large) face mask T-6; (3) the effect of face mask fit by donning a large size face mask T-6, both loose and tightened using thermal manikin, Newton. ANOVA test revealed that the additional N95 middle layer filter has no significant effect on the thermal resistances of all the face masks, and evaporative resistances except for face masks T-2 and T-3 (P-values<0.05) whereas size significantly affected thermal and evaporative resistances (P-values<0.05). The correlation coefficient between the air gap size and the thermal and evaporative resistance of face masks T-6 were $R^2 = 0.96$ and 0.98, respectively. The tight fit large face mask had superior performance in the dissipation of heat and moisture from the skin (P-values <0.05). Three-layer masks incorporating filters and water-resistant and antimicrobial/antiviral finishes did not increase discomfort. Interestingly, using face masks with fitters improved user comfort, decreasing thermal and evaporative resistances in direct opposition to the preconceived notion that safer masks decrease comfort.

## Introduction

Face masks have been recommended to minimize the risk of exposure to a variety of airborne respiratory infectious diseases such as the severe acute respiratory syndrome (SARS) and

**Data Availability Statement:** All relevant data are within the paper and it's Supporting Information files

**Funding:** KG received award FR51906 from Mitacs (https://www.mitacs.ca/en), and award CFPMN1-

026 from the Department of National Defence (https://www.canada.ca/en/department-national-defence/services/contact-us.html). The funders had no role in study design, data collection and analysis, decision to publish, or preparation of the manuscript.

**Competing interests:** The authors have declared that no competing interests exist.

COVID-19. Infectious diseases have significantly increased the use of protective face masks that cover the mouth and nose, to impede the spread of airborne particles generated by infectious patients [1–4]. Face masks also protect uninfected individuals by reducing the wearer's exposure to inhalation of virus droplets/aerosols [1,5–7]. With the recent COVID-19 pandemic, many countries mandated the use of face masks in public places as a potential tool to combat the spread of the virus, following WHO guidelines [8]. The perception of personal discomfort while wearing face masks is cited as a major reason for reluctance towards wearing them, especially the three-layer face masks [2,9–14]. This research was designed to scientifically investigate the validity of this claim.

Face masks could interfere with respiratory and dermal mechanisms of human thermoregulation, which would lead to thermophysiological and sensorial discomfort. Face masks can cause facial heat and moisture accumulation in their microclimate, leading to physiological and psychophysiological strains including local dermal effects, increased inhaled air temperature, core temperature elevation, inspiratory and expiratory resistance, dead air space increase, and a rise in respiratory, heart rates, and both systolic and diastolic blood pressure [10,12,15].

While wearing a face mask, thermal regulation in the human body is regulated through heat exchange between the skin and the environment via the mask. The critical factors determining this thermal balance include the mask's fit, the size of the air gap, and the mask's fabric system [9,16–19]. The head and facial regions play a critical role in thermal regulation and body cooling. Impeding the heat exchange between the facial skin and the ambient environment while wearing a face mask could also disturb skin temperature regulation and cause discomfort [11,20–23].

Wearing face masks, the relative humidity in the air gap caused by human respiration and perspiration can also affect the effective temperature of the air gap [12]. During exhalation, the relative humidity of the microclimate has been reported to increase up to 95%, resulting in an effective temperature rise in the air gap up to 62˚C. This indicates that warm and humid air could increase the intolerance of wearing a face mask [24].

The relative humidity in the air gap caused by human respiration and perspiration results in moisture condensation and affects mask functionality and protective performance. Moisture condensation is undesirable as it causes increased moisture content (retention of water vapor and sweat) within the face mask, and mediates transmission of any infectious agent from the surface of the fabric system to the skin via wicking [25]. In addition, the heat and moisture accumulation within the face mask and the air gap can result in sensorial discomfort leading to intolerance to the mask by the user. A wet and hot fabric may, for instance, cling to the skin and disturb the tactile sensation [9,26].

Furthermore, the relatively high heat and humidity of the expired air and the accumulation of sweat in the mask's microclimate can lead to moisture condensation within its structure, release latent heat of condensation, and disturb thermal regulation [25]. Moisture condensation in face masks can also decrease the air permeability of the textiles due to an increase in the dimensional swelling of the fibers caused by moisture absorption, which can lead to a change in the porosity and thickness of the fabric [27]. This decrease in porosity in turn impacts the air and vapor permeability of the system and impairs heat loss from the air gap to the environment, potentially increasing the breathing resistance [2,9,28–30].

Recent studies on face masks focused on the effect of the fabric system and face mask fit/design on its filtration efficiency [6,7,9]. Multilayer cloth masks were found to have improved filtration capability, blocking as much as 50 to 70% of exhaled small droplets and particles in comparison to single-layer face masks [6]. A two-layer nylon mask with an aluminum nose bridge and a filter insert demonstrated almost two times fitted filtration efficiency than a medical procedure mask with ear loops [19]. Multilayer fabric systems and fabrics with higher

Table 1. Fiber content and structural features of the selected fabrics.

| Fabrics | Fiber content | Fabric Structure | Surface Property |
|---------|---------------|------------------|------------------|
| Fabric A | 65%polyester/ 35%cotton | Plain weave, poplin | Antimicrobial finish (Chlorine) |
| Fabric B | 100%polyester | Double knit interlock | Durable water repellent and antimicrobial and antiviral finish |
| Fabric C | 100%polyester | Double knit interlock | No finish |
| Fabric D | 100%Mulburry silk | Plain weave | No finish |
| Fabric E | 100%cotton | Plain weave, oxford | No finish |
| Fabric F | 100%polypropylene | Nonwoven, meltblown | No finish |
| Fabric G | 100%polypropylene | Nonwoven, meltblown | No finish<br>Filter (NIOSH approved N95)* |
| Fabric H | 50%polyester/ 50%cotton | Plain weave | Poor water repellent surface |
| Fabric I | 100%cotton | Single jersey | No finish |

* A National Institute for Occupational Safety and Health (NIOSH) approved particulate filtering facepiece respirators that filters at least 95% of airborne (particle size 0.1 μm). The experiments were performed by Nelson Labs.

thread counts have also been shown to improve filtration efficiency [6]. Wearing two medical procedure masks has been shown to be more efficient in comparison to donning one mask alone [31]. Adding a filter layer in addition to two layers of cotton or nonwoven fabric improves the filtration efficiency significantly [32]. Further, the filtration capacity of cloth masks is highly dependent on the design, fit, and materials used in the face mask's fabric system. Recent studies have shown that wearing a close-fit face mask can enhance the filtering capability (up to 90%), reduce inhalation of airborne particles, and maximize overall mask performance [19,31,32]. It has also been demonstrated that the fitted filtration efficiency of consumer-grade masks increased from 38.5 to 60.3% when the air loops were tied, and the mask corners were tightened against the wearer's face. Adding mask fitters to two medical procedure masks has further been shown to enhance the masks' performance, demonstrating the effect of fit improvement on filtration efficiency [31].

During the COVID-19 pandemic there have been few systematic studies on adding layers and enhancing face mask fit in favor of increasing a mask's filtration efficiency and face mask protective performance. However, the potential intolerance to wearing these face masks due to the perceived potential thermophysiological strains remains a concern. This research was designed to elucidate the heat and mass transfer mechanisms while wearing commercially available face masks, as well as investigate the effect of fabric properties, fabric structure, and face mask size and fit on its thermophysiological comfort.

## Methods

### Face mask fabric system

Commercially available face masks and fabrics were used and their structure and surface properties are provided in Tables 1 and 2, respectively.

The construction of each fabric system is depicted in Table 2.

### Fabric system property measurements

The physical properties of fabrics including fabric count, mass, thickness and air permeability were measured under standard test conditions (20 ± 2˚C, 65 ± 5% RH), under both dry and wet conditions (Table 2). Fabric count was determined for fabrics A, D, and E used in the structure of face masks D-1, D-4, and D-5, respectively, according to ASTM D3775-17e1 [33].

**Table 2. Physical properties of the fabric systems used in the constructed face masks.**

| Face Mask | Mass (g/m²) | | Thickness (mm) | Density (g/cm³) | | Air permeability (cm³/cm²/s) | |
|---|---|---|---|---|---|---|---|
| | Dry ($M_{conditioned}$) | Wet ($M_{saturated}$) | | Dry | Wet | Dry | Wet |
| (D-1) Fabric A+ Fabric A | 282 | 511 | 0.63 ± 0.0 | 0.45 | 0.81 | 52.2 ± 2.1 | 0 |
| (D-2) Fabric B+ Fabric B | 301 | 458 | 1.08 ± 0.01 | 0.28 | 0.44 | 126.9 ± 1.4 | 98.5 ± 8.2 |
| (D-3) Fabric C+ Fabric C | 284 | 619 | 0.93 ± 0.01 | 0.30 | 0.67 | 172.1 ± 14.3 | 45.0 ± 1.4 |
| (D-4) Fabric D+ Fabric D | 160 | 300 | 0.46 ± 0.01 | 0.35 | 0.66 | 76.1 ± 2.2 | 0 |
| (D-5) Fabric E+ Fabric E | 322 | 704 | 0.88 ± 0.01 | 0.36 | 0.80 | 14.2 ± 0.7 | 0 |
| (T-1) Fabric A+ Fabric G+ Fabric A | 383 | 754 | 0.93 ± 0.01 | 0.41 | 0.81 | 20.0 ± 1.5 | 0 |
| (T-2) Fabric B+ Fabric G+ Fabric B | 391 | 604 | 1.41 ± 0.02 | 0.28 | 0.56 | 25.0 ± 0.3 | 0 |
| (T-3) Fabric C+ Fabric G+ Fabric C | 406 | 928 | 1.26 ± 0.02 | 0.32 | 0.70 | 22.6 ± 0.6 | 0 |
| (T-4) Fabric D+ Fabric G+ Fabric D | 261 | 527 | 0.77 ± 0.01 | 0.34 | 0.68 | 21.3 ± 0.3 | 0 |
| (T-5) Fabric E+ Fabric G+ Fabric E | 423 | 986 | 1.18 ± 0.02 | 0.36 | 0.79 | 11.9 ± 0.2 | 0 |
| (T-6) Fabric E+ Fabric F+ Fabric E | 361 | 774 | 1.17 ± 0.02 | 0.31 | 0.66 | 14.6 ± 0.1 | 0 |
| (Q-1) Fabric H+ Fabric I+ Fabric I+ Fabric I | 915 | 2050 | 2.1 ± 0.0 | 0.44 | 0.99 | 15.9 ± 0.3 | 0 |

The conditioned mass ($M_{conditioned}$) of each fabric was measured according to ASTM D3776/D3776M-20 and is presented as grams per unit area (g/m²) [34]. The fabric thickness was measured according to ASTM D1777-96(2019) [35] and the fabric air permeability according to ASTM D737-18 [36]. The density of the fabric (g/cm³) was determined by dividing the mass of the fabric (g/m²) by the thickness (m) and applying a conversion factor of 1000. Mass, thickness and air permeability of the fabric systems were also measured in the wet condition ($M_{saturated}$). For wet tests, the conditioned specimens were immersed in distilled water for at least 5 minutes, taken out, and squeezed to remove the excess liquid water and free water using commercial blotting paper. The specimens were then sealed in plastic bags and allowed to condition for 12 hours. The specimens were subsequently removed from the sealed bag and weighed to measure $M_{saturated}$.

## Face mask construction

Face masks were constructed using the fabric systems listed in Table 1 and assembly code in Table 2, as per the measurements given in Table 3 and Fig 1A. Fig 2 also shows the fabric systems and the face masks constructed for this study. Face masks D-1 to D-5 were two-layer commercially available face masks. To investigate the effect of an additional N95 filter (fabric G, Table 1) on the thermophysiological comfort of the commercially available face masks (D-1 to D-5), face masks T-1 to T-5 were constructed as three-layer face masks. To construct these, the additional nonwoven middle layer filter was added to the two-layer fabric systems D-1 to D-5. The effect of an N95 and a typical filter (fabric F) on the thermophysiological comfort

Table 3. Face mask's dimensions, area, and air gap size.

| Face mask measurement | Size | | |
|---|---|---|---|
| | Small | Medium | Large |
| A-B (mm) | 19 | 25 | 25 |
| B-C (mm) | 70 | 86 | 114 |
| C-D (mm) | 25 | 32 | 38 |
| D-E (mm) | 64 | 102 | 102 |
| E-F (mm) | 70 | 70 | 89 |
| F-A (mm) | 76 | 98 | 114 |
| G (mm) | 83 | 108 | 120 |
| Elastic Earloop (mm) | 127 | 140 | 152 |
| Area ($cm^2$) | 142 | 223 | 306 |
| Air gap size ($cm^3$) | 124 | 499 | 816 |

was investigated in face masks T-5 and T-6, which have the same outermost and innermost layer but different middle layer filters. Face masks T-2 and T-3 have fabrics with the same fiber content, yarn type and fabric structure in their outermost and innermost layers but different finishes. Face mask T-2 has fabric B, which included durable water repellent and antimicrobial and antiviral finishes, whereas T-3 has no surface finish (Fabric C). This set was chosen to investigate the impact of finishing on thermophysiological face mask comfort.

## Face mask size, area and air gap measurements

While performing the comparison of face masks with different sizes and fit, the covered area of the face zone and the air gap size between the mask and the face zone could alter thermal and evaporative resistances and needed to be taken into account. To determine the effect of size, face mask T-6 was tailored into small, medium and large size variants (Table 3). The air gap size of T-6 for these three sizes was also estimated by volume of a triangular prism (Fig 1B) with the values for "G" and "a" in Table 3. The value of "b" was assumed to be the width of the manikin's face and was kept constant (160 mm) for all face mask sizes. To determine the effect of fitting, the large size face mask T-6 was tested at three fits: loose, normal, and tight. For the tight fit, the face mask was tightened to the face by placing a ring of three-ganged rubber bands over the mask, with the center rubber band over the nose and chin of the manikin and the right and left sides looped over the 3D-printed ear guard. The latter donning approach was employed as proposed by Calpp et al. [19] where they found the fitted approach would significantly enhance the filtration efficiency of medical procedure masks from 38.5 to 78.2% in a study conducted on human participants.

## Thermal manikin

Thermal and evaporative resistances are critical to predict the fabric systems' ability to maintain thermophysiological regulation [11,16]. This study was conducted in a series of experiments carried out using a sweating thermal manikin, Newton, in a climatic chamber (Thermetrics LLC, Seattle, WA). The sweating manikin was a standard 26-thermal zone configuration designed to produce an accurate, repeatable measurement of the thermal and evaporative resistance of garments under steady-state conditions. Newton is a 50th percentile male fitted with heaters, temperature sensors and sweating nozzles. The thermal resistance and evaporative resistance of the face masks were measured in accordance with Test Methods ASTM F1291 and ASTM F2370, respectively.

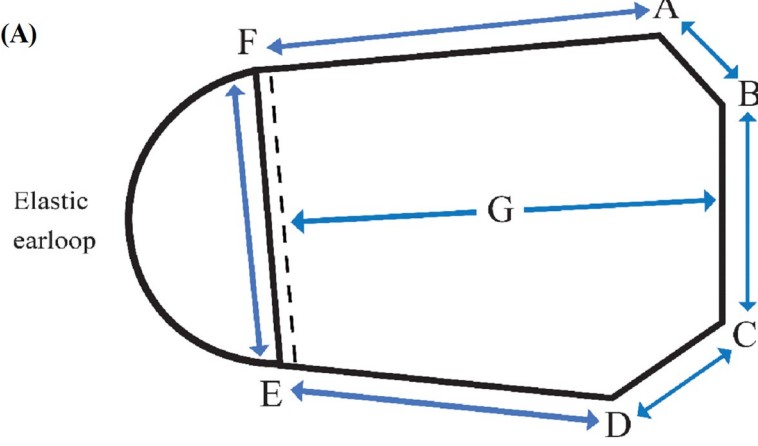

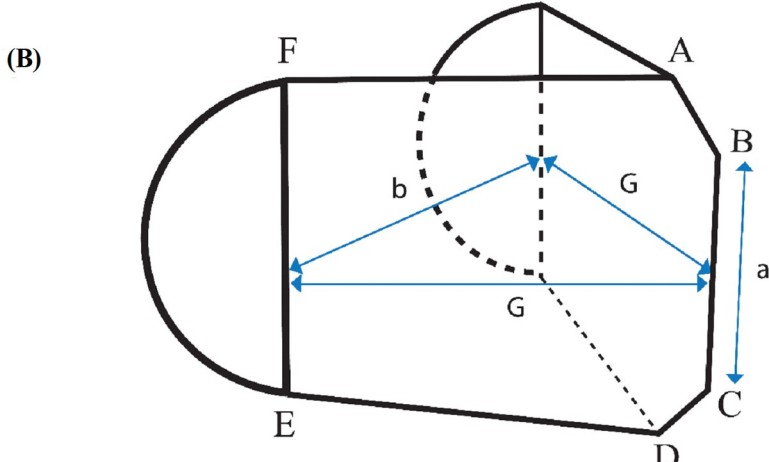

**Fig 1.** Face mask's dimensions for (A) size measurement and (B) air gap measurements.

## Test protocol

To assess the thermophysiological comfort of the face masks, three approaches were employed: (1) the evaporative and thermal resistances were measured in all the test face masks using the medium size to determine the effect of fabric properties; (2) the effect of face mask size was investigated by testing close-fitted (small), fitted (medium) and loose fitted (large) face mask T-6; (3) the effect of face mask fit was studied by donning a large size face mask T-6, both loose and tightened. As this study was intended to determine thermal and evaporative resistance changes caused solely by the face mask, only the face zone of the mannequin ($0.0475$ m$^2$) covered by face mask was considered. For the comparison of the face masks with similar size and design, the air gap size and the covered face zone area were kept constant in order to isolate the effect of the fabric properties on the evaporative and thermal resistances of the face mask. When comparing the face mask with different sizes and fit, the covered face zone area and the air gap size were different and affected the predicted thermal and evaporative resistances.

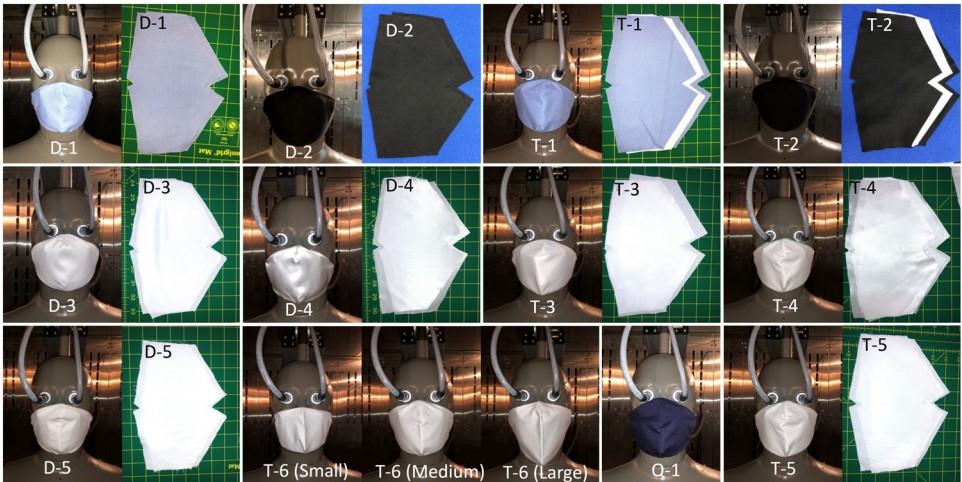

**Fig 2. Face masks and their fabric systems.**

For the wet tests, sweat rates were set at 1500 (ml/h.m$^2$) for face and head zones to achieve 100% RH on the skin surface. It was previously reported that moist expired air increases the relative humidity of the air gap up to $\approx$ 90%, disturbs thermal sensation, and increases skin and mask wetness [24]. The predefined sweat rate was assumed to compensate for the relative humidity within the air gap caused by the moist expired air. The environmental conditions were simulated in the climatic chamber for the dry tests at an ambient temperature of $t_a$ = 20°C, a relative humidity of RH = 65%, and an air velocity of 0.5 m/s. Each test was replicated three times, and each test was continued until a steady-state condition was achieved according to ASTM F1291.

## Statistical analysis

Comparisons between two groups were performed by using unpaired Students' t-test assuming a two-tailed distribution and unequal variances. For multiple comparisons, ANOVA was applied. Any P-values less than 0.05 were deemed to be insignificant, i.e. for P-values that did not span 95% of the confidence interval, the property was inferred to be statistically significant and to affect the thermal and evaporative resistances of the face masks. The relation between the dependent (evaporative resistance and thermal resistance) and the independent (face mask properties) variables were obtained using Pearson correlation. Normality assumption was checked for all parametric tests.

## Results

### Thermal and evaporative resistances of the face masks

Our thermal and evaporative resistance analyses showed that wearing a face mask causes impediments to heat and moisture transfer (Table 4). The thermal resistances of the tested face masks and the surface air layer of the face zone (Rct) were measured in the range of 0.084 to 0.089 m$^2$°C/W, while the thermal resistance of the air layer on the surface of the nude manikin (Rct$_0$) was 0.064 m$^2$°C/W. These values account for a total impediment in heat dissipation of 30–40%. The evaporative resistance analysis for the nude (Ret$_0$ = 10.19 Pa.m$^2$/W) and the manikin with face masks and the surface air layer of the face zone (Ret = 12.62–16.46 Pa.m$^2$/

**Table 4. Thermal resistance, Rcf, and evaporative resistance, Ref, of the various mask fabric assemblies.** One standard deviation (SD) is reported in parentheses.

| Face mask fabric system (assembly code) | Size | Fit | Sweating Manikin | |
|---|---|---|---|---|
| | | | Rcf*×$10^3$ (m²·C/W) (SD) | Ref** (Pa.m²/W) (SD) |
| (D-1) Fabric A+ Fabric A | Medium | Normal | 22.33 (1.25) | 2.70 (0.08) |
| (D-2) Fabric B+ Fabric B | | | 21.67 (0.58) | 4.52 (0.06) |
| (D-3) Fabric C+ Fabric C | | | 21.00 (0.82) | 3.92 (0.14) |
| (D-4) Fabric D+ Fabric D | | | 20.00 (1) | 2.43 (0.33) |
| (D-5) Fabric E+ Fabric E | | | 22.00 (0) | 3.01 (0.42) |
| (T-1) Fabric A+ Fabric G+ Fabric A | | | 24.67 (0.58) | 3.31 (0.13) |
| (T-2) Fabric B+ Fabric G+ Fabric B | | | 23.00 (0) | 6.27 (0.46) |
| (T-3) Fabric C+ Fabric G+ Fabric C | | | 22.67 (0.58) | 5.02 (0.25) |
| (T-4) Fabric D+ Fabric G+ Fabric D | | | 22.00 (1) | 2.93 (0.09) |
| (T-5) Fabric E+ Fabric G+ Fabric E | | | 23.50 (1.29) | 3.51 (0.10) |
| (T-6) Fabric E+ Fabric F+ Fabric E | Small | Normal | 18.00 (0.32) | 1.83 (0.11) |
| | Medium | | 20.50 (0.71) | 2.94 (0.35) |
| | Large | | 26.33 (0.38) | 4.91 (0.34) |
| | Large | Tight | 23.00 (0) | 3.75 (0.13) |
| | | Normal | 26.33 (0.38) | 4.91 (0.34) |
| | | Loose | 29.60 (1.05) | 6.56 (0.31) |
| (Q-1) *** Fabric H+ Fabric I+ Fabric I+ Fabric I | Medium | Normal | 23.50 (0.59) | 4.04 (0.10) |

\* Thermal resistance (Rcf) values of the face mask are determined by subtracting the total thermal resistance (Rct) of the face mask and the air layer of the face zone from the air layer resistance on the surface of the nude manikin in dry mode (Rct$_0$), (assuming that the boundary layer of the nude manikin and the boundary layer of the clothed manikin are equal in dry mode).

\*\* Evaporative resistance (Ref) values of the face mask are determined by subtracting the total evaporative resistance (Ret) of the face mask and the air layer of the face zone from the air layer resistance on the surface of the nude manikin in sweating mode (Ret$_0$), (assuming that the boundary layer of the nude manikin and the boundary layer of the clothed manikin are equal in sweating mode).

\*\*\* The particle filtration efficiency (PFE) of fabric system Q-1 was assessed and resulted in an average filtration efficiency of 94% (filters at least 94% of airborne particles, particle size 0.1 μm). The experiments were performed by Nelson Labs.

W) revealed that wearing any face mask causes a resistance to moisture transfer by approximately 25 to 60%.

### Effect of fabric properties on the total thermal and evaporative resistance

The thermal and evaporative resistances of the two-layer face masks were in the range of $20\times10^{-3}$ to $22.33\times10^{-3}$ m²·C/W and 2.43 to 4.52 Pa.m²/W, respectively (Fig 3A and 3B). Among the studied two-layer face masks, the 100%silk face mask (D-4) exhibited the lowest thermal and evaporative resistances. The excellent heat and moisture transfer properties of silk, and its relatively lower mass and thickness among the fabric systems, caused more dissipation of thermal energy and moisture to the environment.

Water-resistance is important to consider within a mask structure as it decreases surface wetting and in-depth moisture penetration. To analyze the effect of the water-resistance of face masks on their moisture transfer, the density of the fabrics was measured in both wet and dry conditions (Table 2). There was a negative correlation between physical properties of the fabric systems (density of the wet fabric systems) and the evaporative resistance of the face mask layer ($R^2$ = 0.80).

Analyses of the air permeability of the two-layer fabric systems showed that the air permeability decreased significantly in the wet condition. The air permeability measurements of

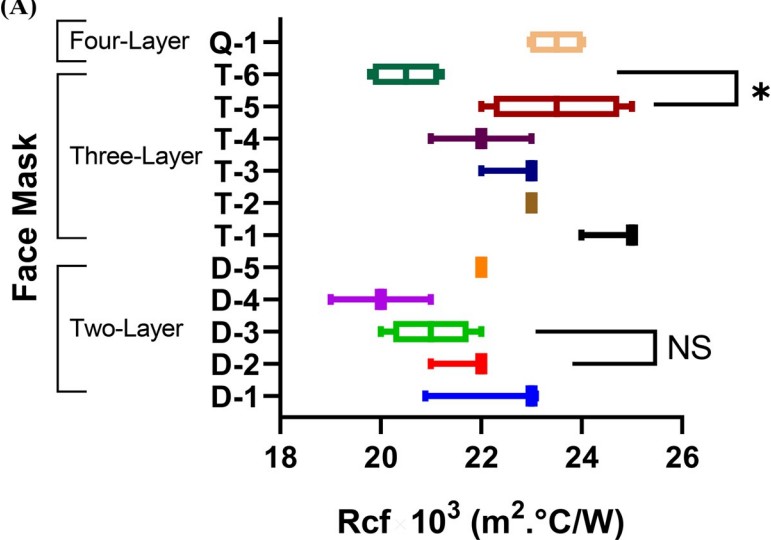

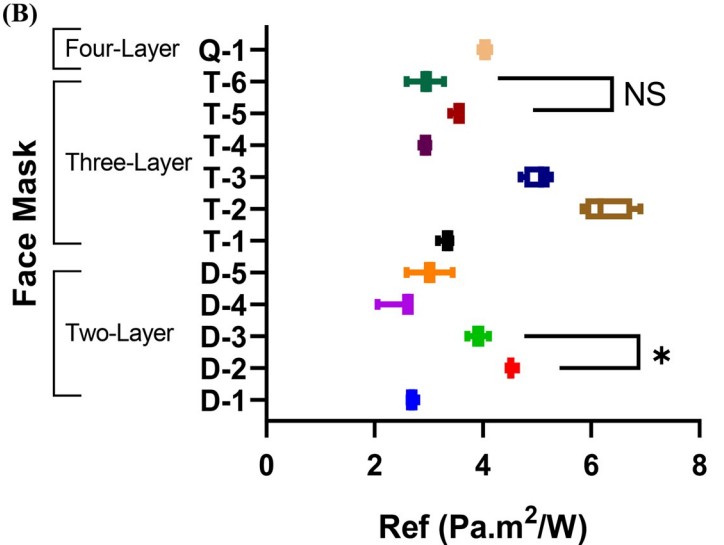

**Fig 3.** (A) Thermal and (B) evaporative resistances of the various face mask assemblies (* = P <0.05; NS = not statistically significant).

woven fabric systems D-1, D-4, and D-5 all reduced to 0, from 52, 76 and 14 ($cm^3/cm^2/s$) respectively. The fabrics used in the structure of fabric systems D-1, D-4, and D-5 exhibited closely woven structures with fabric A = 40×20, fabric D = 55×50, and fabric E = 40×40 yarn/cm. The small fabric pores resulted in clinging of moisture to the fibers and blocking the airways within their structure. When the fibers in the structure of these fabrics absorb water, they may swell transversely and axially [37]. As such, the pores of the closely woven fabrics can be completely blocked making the fabric impermeable to air. The air permeability of the knitted fabric systems D-2 and D-3 also decreased by ≈ 40 and 70%, respectively, in the wet condition. Knitted fabric B and fabric C have the same fabric structure and were used in face masks D-2 and D-3, respectively. These fabrics have a larger pore size in comparison to the woven fabrics

used in this study and had less air blockage. In addition, these fabrics are made of polyester and exhibited less dimensional change compared to the other test fabrics comprised of natural fibers. The air permeability of fabrics D-2 and D-3 decreased significantly, but was retained in the wet condition. This suggested that the use of knitted fabric structures in face masks could reduce breathing resistance for the wearer in the wet condition. However, fabric D-2 was 75% more air-permeable than fabric D-3 in the dry condition. The considerably lower air-permeability of D-2 in the wet condition was attributed to the durable water repellent finish. Fabric B's inherently greater hydrophobicity in fabric system D-2, compared to fabric C in D-3, resulted in less adhesion of moisture in D-2, allowing for greater air permeation. This demonstrated that water repellent finishing can also improve the thermophysiological comfort of masks.

## Effect of fabric structure on the total thermal and evaporative resistance

The effect of an additional layer added to the face mask on the thermophysiological comfort was analyzed in three-layer masks T-1, T-2, T-3, T-4 and T-5, each with an additional N95 (NIOSH approved) nonwoven filter middle layer. The additional N95 middle layer filter increased thermal resistance 5 to 10% and evaporative resistance 15 to 35% (Fig 3A and 3B). Multiple comparisons between the evaporative and thermal resistances of the face masks (ANOVA) revealed that the additional N95 middle layer filter has a statistically insignificant effect on the thermal resistances of all the face masks, as well as their evaporative resistances except for face masks T-2 and T-3 (P-values<0.05). The decrease in the air permeability of T-2 and T-3 in the wet condition explains why the additional N95 filter affected moisture transfer in the face masks. Considering the N95 filtration efficiency, it can be inferred that these three-layer face masks blocked small droplets and particles from passing through their fabric system. The impact on thermal and evaporative resistance of an N95 versus a typical filter (Fabric F) can be compared using face masks T-5 and T-6. Face mask T-5 (N95 middle layer) exhibited only 15% higher heat resistance and 20% higher moisture resistance, in comparison to face mask T-6 with the typical nonwoven middle layer (Fig 3A and 3B). This demonstrated that using an N95 filter offers comparable comfort to masks with a nonwoven filter, but offers enhanced protection to the user.

The dry mass of the nonwoven N95 filter (fabric G) was 102 g/m$^2$ and 247 g/m$^2$ in the wet condition. A 12 to 20% increase in water retention was observed in the face masks with the additional hydrophilic nonwoven N95 filter without any water-resistance surface finishing (T-1, T-3, T-4, and T-5), compared to their corresponding two-layer face masks (D-1, D-3, D-4 and D-5). However, using fabric B (durable water repellant finish) significantly reduce wettability and blocked the penetration of moisture to its underlying N95 nonwoven filter. Face mask T-2, with the additional hydrophilic nonwoven N95 filter within the durable water-repellent fabrics, had a minor effect on its water retention (5% increase) in comparison to face mask D-2 (no filter). The moisture resistance offered by the durable water repellent finish resulted in the highest evaporative resistance being observed in face mask T-2 (6.27 Pa.m$^2$/W).

Four-layer face mask Q-1 utilized a 50%cotton/50%polyester woven fabric as a face fabric (Fabric H) with three underlying knitted 100%cotton layers (Fabric I). This fabric construction has been commercially used as a reusable N94 filter for respirators. Here, the Q-1 fabric system was made into a face mask to see the effect of multiple layering on the evaporative and thermal resistance of the face mask. The thermal resistance of the four-layer mask was measured as 23.5×10$^{-3}$ m$^2$°C/W, similar to the average thermal resistance of the three-layer (22.7×10$^{-3}$ m$^2$°C/W) and two-layer (21.4 m$^2$°C/W) face masks (Table 4). The average values of evaporative resistance for the three-layer masks (3.13 Pa.m$^2$/W) and the two-layer masks (4 Pa.m$^2$/W)

indicated that the moisture transfer of the four-layer face mask Q-1 (4.04 Pa.m$^2$/W) was comparable. The single jersey knitted structure of fabric I had larger pores in its structure in comparison to the nonwoven and woven fabrics, which allowed air and moisture transfer through its structure while still offering N94 filtration efficiency.

### Effect of size and fit on the total thermal and evaporative resistance

The covered surface area and air gap sizes could interfere with thermoregulation and reduce the speed at which the body loses heat and moisture. For this purpose, face mask T-6 was made into a size small, medium, and large, and the evaporative and heat resistances were determined (Fig 4A and 4B).

The thermal and evaporative resistances of face mask T-6 in different sizes were influenced by the size of the microclimate and the air gap between the mask and the human body. Multiple comparisons between the evaporative and thermal resistances of the face mask T-6 in different sizes (ANOVA) revealed that a change in size has a statistically significant effect on the thermal and evaporative resistances (P-values<0.05). The correlation coefficient between the air gap size and the thermal and evaporative resistances of face masks T-6 were R2 = 0.96 and 0.98, respectively. The thermal resistance (from $18\times10^{-3}$ to $26.33\times10^{-3}$ m$^2$°C/W) and evaporative resistance (from 1.83 to 4.91 Pa.m$^2$/W) of face mask T-6 increased as the size changed from small to large. The covered area of face mask T-6 also had a significant impact on the thermal and evaporative resistances of the face mask, with R2 = 0.98 and R2 = 0.99 for thermal and evaporative resistances, respectively. A larger surface area is covered when, for example, a medium-sized individual wears a larger face mask. The covered area of the face mask increased 35% as the size changed from medium to large, resulting in 30 and 65% increases in thermal and evaporative resistances, respectively.

The increased air gap in a loose fit large face mask (Figs 4A, 4B and 5B) decreased heat and moisture transfer from the skin to the environment and imposed an increased heat burden, increasing thermophysiological discomfort. The tight fit large face mask, shown in Fig 5B and 5C, had superior performance in the dissipation of heat and moisture from the skin (P-values <0.05). The decrease in the air gap size of the loose fit large face mask T-6, by using the three-rubber bands method, decreased thermal and evaporative resistances 22 and 42%, respectively (Fig 4A and 4B).

### Discussion

The total evaporative and thermal resistance of the face mask can be explained as a series of heat and moisture flows from the skin to the environment, as illustrated in Fig 6. The thermal and evaporative resistance within the face mask air gap is affected by face mask size, fit, properties, and structure of the fabric system comprising the face mask. In addition, the thermal (0.064 m$^2$°C/W) and evaporative (10.19 Pa.m$^2$/W) resistances of the air layer can also contribute to total evaporative and thermal resistances, and was accordingly kept constant in this study.

Our data shows surface finishing had a minor effect on the heat and moisture transfer of the face masks. Face masks D-2 and D-3 employed similar fabrics (100% polyester, double-knit interlock) in their outer and inner layers, but face mask D-2 had a durable water repellent and antimicrobial/antiviral finish, while D-3 had no surface finishing. The similar values of thermal and evaporative resistances obtained for D-2 and D-3 showed that surface finishing had no significant impact on heat transfer and a slight decrease in moisture transfer (Fig 3A and 3B). Therefore, a face mask with water-resistant surface properties to resist surface

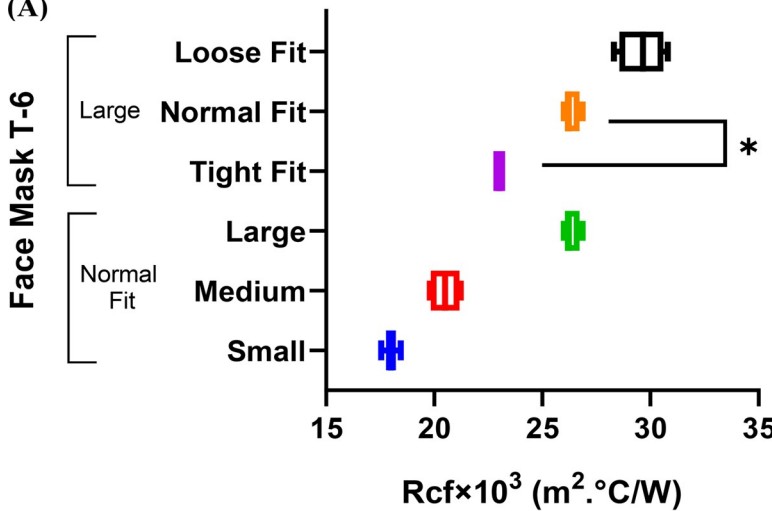

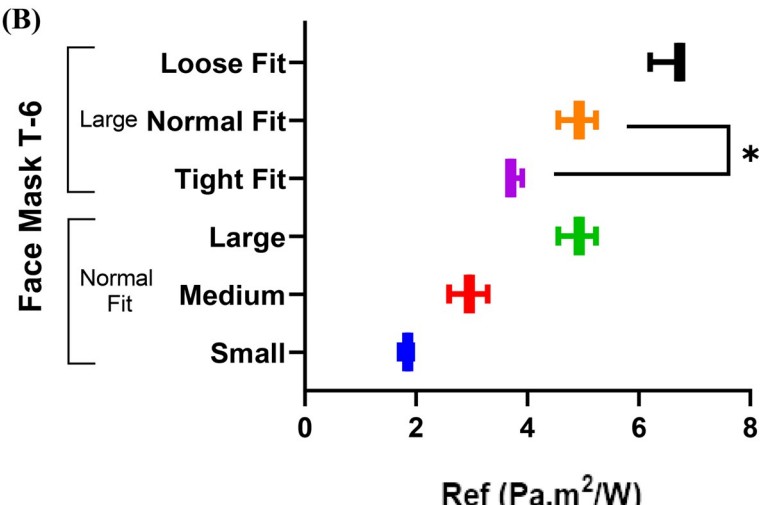

**Fig 4.** Face mask T-6's (A) thermal (B) evaporative resistances (* = P <0.05).

wetting, as well as antimicrobial/antiviral properties to inactivate bacteria and viruses, would be desirable for enhanced protection and will not sacrifice comfort.

The data shows considerably less moisture in the D-2 fabric system with less moisture transfer compared to the other two-layer masks tested. The durable water repellent finish of the D-2 face mask resisted wetting by blocking water penetration through the fabric system, thereby enhancing evaporative resistance. This could also block the transfer of virus-containing expired moist air from the inner to outer layer of the face mask. In face masks D-1, D-3, D-4, and D-5 (no finish) the wet expired air can accelerate the wetting process and enhanced moisture penetration, which could result in viral penetration by water diffusion or bulk fluid motion through the fabric systems' capillaries.

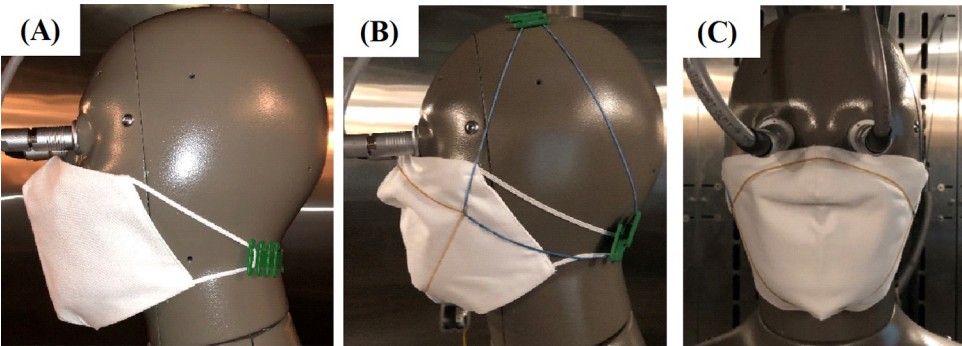

**Fig 5.** Large face mask T-6 (A) loose fit and (B) and (C) tight fit (three rubber bands method [19].

Air permeability studies have shown that an increase in the moisture content of the face mask could also reduce its ability to pass air, affecting thermophysiological comfort. Interestingly, our data on water repellent finishing demonstrated an improvement in the thermophysiological comfort of masks. This is supported by the previous studies which have shown that moisture condensation and accumulation within the fabric structure could decrease the air permeability of the face mask and potentially increase the breathing resistance [2,28]. This in turn can impair heat loss to the environment, causing thermophysiological discomfort.

The increase in water retention observed in the face masks with the additional hydrophilic nonwoven N95 filter without any water-resistance surface finishing (T-1, T-3, T-4, and T-5), compared to their corresponding two-layer face masks (D-1, D-3, D-4 and D-5) could be explained by moisture transfer mechanism. The transfer of moisture from the hydrophilic surfaces of fabrics A, C, D and E to the nonwoven filter. The increased retention of water vapor and sweat could also increase the transmission of an infectious agent from the outer layer to

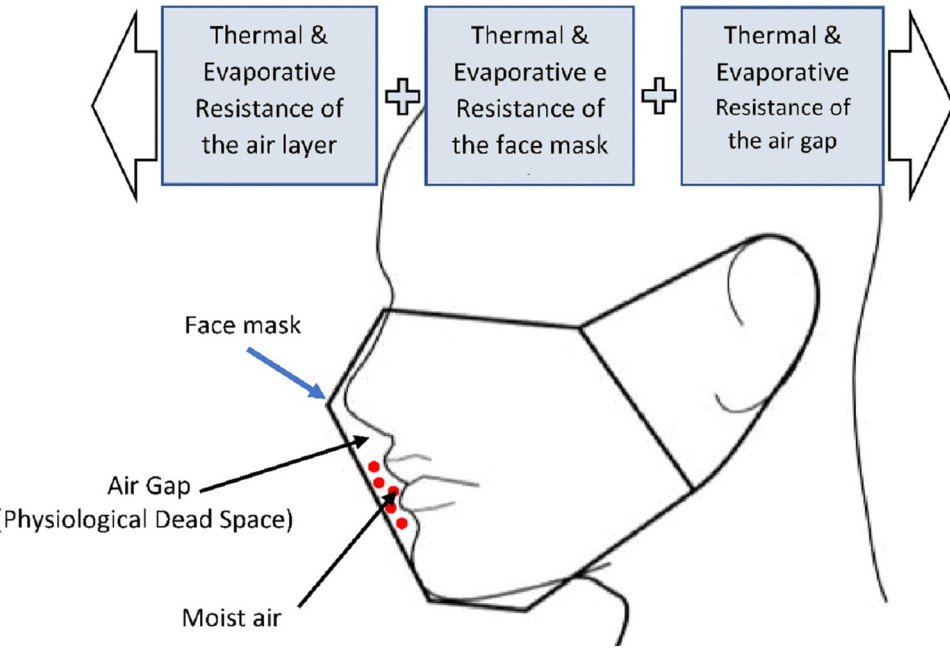

**Fig 6. Schematic illustration of the air gap and moist expired air.**

the inner layer or vice versa, via wicking [25]. Moreover, this phenomenon could in turn increase the diffusion velocity of liquid water and compromise face mask filtration efficiency.

The increase in thermal resistance and evaporative resistances of the mask with respect to its size change from small to large could be attributed to the increased heat and moisture accumulation in the larger air gap compared to smaller-sized face mask. A thin layer of air, having ~8X lower thermal conductivity than the fabric fibers, made the system act as an insulator. In thin air gaps, air and water vapor cannot circulate between the body and the environment, limiting moisture and heat dissipation and thereby increasing evaporative and thermal resistance [26,30,38,39]. This heat and moisture accumulation in the air gap could significantly affect facial temperature regulation, leading to thermal discomfort [11,40]. The air gap can be considered as the dead air space between the face mask and the human face (Fig 6). Overall, our results demonstrate using a three-layer masks with filters and water-resistant and antimicrobial/antiviral finishes have no negative impact on thermophysiological comfort. However, in contrast using face masks with fitters improved user comfort by decreasing thermal and evaporative resistances.

## Conclusion

Despite the superior protective performance of multilayer fitted face masks, the public are often reluctant to wear them, citing discomfort associated with their usage. This research discourages this assumption by demonstrating that multilayer face masks, fitted face masks, and face masks with durable water repellent and antiviral/antimicrobial finishes in their inner and outer fabric structure show no or little effect on thermophysiological comfort properties. Analysis of the studied face masks' thermal resistance and evaporative resistance revealed that adding an N95 filter to a two-layer fabric system has only a minor effect on the face masks' heat and moisture transfer properties. Although large face masks can increase thermal and evaporative resistances, a large fitted mask can reduce the impediment to heat and moisture transfer by reducing the air gap size. As such, mask fitters are recommended to enhance the heat and moisture transfer properties of face masks. Overall, users are recommended to wear safer face masks as they do not result in a meaningful decrease in thermophysiological comfort.

## Supporting information

**S1 Data.**
(XLSX)

## Acknowledgments

The authors also thank Adrian Bussoli and Mohamed Packir (Alberta Innovation Center), and Steve Bommer (Spirit West) for providing fabrics and technical support throughout this work.

## Author Contributions

**Conceptualization:** Farzan Gholamreza, Anupama Vijaya Nadaraja, Kevin Golovin.

**Formal analysis:** Farzan Gholamreza.

**Funding acquisition:** Kevin Golovin.

**Methodology:** Farzan Gholamreza, Anupama Vijaya Nadaraja, Abbas S. Milani, Kevin Golovin.

**Supervision:** Abbas S. Milani, Kevin Golovin.

**Writing – original draft:** Farzan Gholamreza.

**Writing – review & editing:** Farzan Gholamreza, Anupama Vijaya Nadaraja, Abbas S. Milani, Kevin Golovin.

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
