## [Editor Report · Decision Letter 0]

7 Oct 2021

PONE-D-21-22455Enhanced protection face masks do not adversely impact thermophysiological comfortPLOS ONE

Dear Dr. Golovin,

Thank you for submitting your manuscript to PLOS ONE. After careful consideration, we feel that it has merit but does not fully meet PLOS ONE’s publication criteria as it currently stands. Therefore, we invite you to submit a revised version of the manuscript that addresses the points raised during the review process.

We look forward to receiving your revised manuscript.

Kind regards,

Manpreet Singh Bhatti, B.E. (Civil), M.E. (Env. Eng.), Ph.D.

Academic Editor

PLOS ONE

Journal Requirements:

The authors thank The University of British Columbia Okanagan for financial support through the Eminence program. This work was partially funded by Mitacs Accelerate, through grant FR51906, and by the Department of National Defence under contract CFPMN1-026.  The authors also thank Adrian Bussoli and Mohamed Packir (Alberta Innovation Center), and Steve Bommer (Spirit West) for providing fabrics and technical support.

KG received award FR51906 from Mitacs (https://www.mitacs.ca/en), and award CFPMN1-026 from the Department of National Defence (https://www.canada.ca/en/department-national-defence/services/contact-us.html). The funders had no role in study design, data collection and analysis, decision to publish, or preparation of the manuscript.

This work was partially funded by Mitacs Accelerate, through grant FR51906

Additional Editor Comments:

The authors studied 12 different layered commercial facemasks and analysed the evaporative and thermal resistances using sweating Manikin. At first instance, the manuscript seems interesting, but after reading the complete manuscript, it lacks coherence. There are several variables (different materials, different layers) to check the thermophysiological comfort. My observations are as under:

Out of 12 combinations, only one is significant at p<0.05 and only one is not significant (D-2 vs. D-3). What about others.

The introduction section may be condensed by 30-50%.

Add latest references as the majority of references are more than 5-year-old.

Multiple comparison tests revealed few interactions. Authors may try to delete non-significant combinations.

Any logic of combining 3 different masks T-1 to T-6 and Q-1.
---

## [Decision Letter · Decision Letter 1]

19 Jan 2022

PONE-D-21-22455R1Enhanced protection face masks do not adversely impact thermophysiological comfortPLOS ONE

Dear Dr. Golovin,

Thank you for submitting your manuscript to PLOS ONE. After careful consideration, we feel that it has merit but does not fully meet PLOS ONE’s publication criteria as it currently stands. Therefore, we invite you to submit a revised version of the manuscript that addresses the points raised during the review process. Please see the reviewer comments below.

We look forward to receiving your revised manuscript.

Kind regards,

Hanna Landenmark

Senior Editor, PLOS ONE

on behalf of 

Yasir Nawab

Journal Requirements:

Additional Editor Comments (if provided):

Reviewers' comments:

Reviewer's Responses to Questions

**Comments to the Author**

1. If the authors have adequately addressed your comments raised in a previous round of review and you feel that this manuscript is now acceptable for publication, you may indicate that here to bypass the “Comments to the Author” section, enter your conflict of interest statement in the “Confidential to Editor” section, and submit your "Accept" recommendation.

Reviewer #1: All comments have been addressed

Reviewer #2: (No Response)

2. Is the manuscript technically sound, and do the data support the conclusions?

Reviewer #1: Yes

Reviewer #2: Yes

3. Has the statistical analysis been performed appropriately and rigorously? 

Reviewer #1: Yes

Reviewer #2: Yes

4. Have the authors made all data underlying the findings in their manuscript fully available?

Reviewer #1: Yes

Reviewer #2: Yes

5. Is the manuscript presented in an intelligible fashion and written in standard English?

Reviewer #1: Yes

Reviewer #2: (No Response)

6. Review Comments to the Author

Reviewer #1: The article titled “Enhanced protection face masks do not adversely impact thermophysiological comfort” deals with the development of mask with enhanced physiological comfort and other more protection. The focus of this study is to development and analysis of different fabric combinations in the face mask.

All the comments of the previous reviewer were addressed properly and explained according to need.

Here are some points in the current embodiment, need to address for better understanding of the research.

1. You have selected the fabric H and Fabric I but not used in the final fabric selection of fabric face mask development. Why?

2. What is the reason of higher AP of D-3 as compared to D-4 in wet condition as both have same fabric structure?

3. Please provide the layer-to-layer configuration of the masks and pictures of original developed samples.

4. Overall research is interesting but needs to address the above points for further clarification.

This is an interesting and novel research work which is very productive for the subjected field.

Reviewer #2: (No Response)

7. PLOS authors have the option to publish the peer review history of their article (what does this mean?). If published, this will include your full peer review and any attached files.

Reviewer #1: **Yes: **Raja Muhammad Waseem Ullah Khan

Reviewer #2: No

---

## [Author Response · Author response to Decision Letter 1]

27 Jan 2022

Hanna Landenmark

Senior Editor, PLOS ONE

Dear Dr. Landenmark,

Thank you for sending us the reviewer comments. They were very helpful and we’ve improved the manuscript in light of the reviewer concerns and comments. Please see our responses below response in blue font. The original comments have been reproduced in black font.

Q: You have selected the fabric H and Fabric I but not used in the final fabric selection of fabric face mask development. Why?

A: Fabrics H and I were used in the construction of face mask Q-1. This was mentioned in Table 4 but was incorrectly labeled in Table 2, which has been corrected in the updated manuscript. 

Q: What is the reason of higher AP of D-3 as compared to D-4 in wet condition as both have same fabric structure?

A: We believe the reviewer is referring to fabrics D-2 and D-3 as they have the same fabric structure unlike D-3 and D-4. The reason for the higher AP of D-2 as compared to D-3 is that D-2 has the Durable Water Repellent finish whereas D-3 does not. As we state on page 13-14:

“The air permeability of fabrics D-2 and D-3 decreased significantly, but was retained in the wet condition. This suggested that the use of knitted fabric structures in face masks could reduce breathing resistance for the wearer in the wet condition. However, fabric D-2 was 75% more air-permeable than fabric D-3 in the dry condition. The considerably lower air-permeability of D-2 in the wet condition was attributed to the durable water repellent finish. Fabric B’s inherently greater hydrophobicity in fabric system D-2, compared to fabric C in D 3, resulted in less adhesion of moisture in D-2, allowing for greater air permeation. This demonstrated that water repellent finishing can also improve the thermophysiological comfort of masks.”

Q: Please provide the layer-to-layer configuration of the masks and pictures of original developed samples. 

A: We have now added Fig 2 which represents the face masks used in the study and their fabric systems. The configuration of layers is now provided in Tables 2 and 4.

---

## [Decision Letter · Decision Letter 2]

24 Feb 2022

Enhanced protection face masks do not adversely impact thermophysiological comfort

PONE-D-21-22455R2

Dear Dr. Golovin,

We’re pleased to inform you that your manuscript has been judged scientifically suitable for publication and will be formally accepted for publication once it meets all outstanding technical requirements.

Kind regards,

Yasir Nawab, PhD

Academic Editor

PLOS ONE

Additional Editor Comments (optional):

Reviewers' comments:

Reviewer's Responses to Questions

**Comments to the Author**

1. If the authors have adequately addressed your comments raised in a previous round of review and you feel that this manuscript is now acceptable for publication, you may indicate that here to bypass the “Comments to the Author” section, enter your conflict of interest statement in the “Confidential to Editor” section, and submit your "Accept" recommendation.

Reviewer #1: All comments have been addressed

Reviewer #2: All comments have been addressed

2. Is the manuscript technically sound, and do the data support the conclusions?

Reviewer #1: Yes

Reviewer #2: Yes

3. Has the statistical analysis been performed appropriately and rigorously? 

Reviewer #1: Yes

Reviewer #2: Yes

4. Have the authors made all data underlying the findings in their manuscript fully available?

Reviewer #1: Yes

Reviewer #2: Yes

5. Is the manuscript presented in an intelligible fashion and written in standard English?

Reviewer #1: (No Response)

Reviewer #2: Yes

6. Review Comments to the Author

Reviewer #1: (No Response)

Reviewer #2: The manuscript provided a deep insight into thermophysiological comfort properties of face masks and is recommended for publication.

7. PLOS authors have the option to publish the peer review history of their article (what does this mean?). If published, this will include your full peer review and any attached files.

Reviewer #1: **Yes: **Raja Muhammad Waseem Ullah Khan

Reviewer #2: **Yes: **Asfandyar Khan

---

## [Editor Report · Acceptance letter]

30 Mar 2022

PONE-D-21-22455R2 

Enhanced protection face masks do not adversely impact thermophysiological comfort 

Dear Dr. Golovin:

I'm pleased to inform you that your manuscript has been deemed suitable for publication in PLOS ONE. Congratulations! Your manuscript is now with our production department. 

Kind regards, 

on behalf of

Dr. Yasir Nawab 

Academic Editor

PLOS ONE